# Differential Impact of Tumor Endothelial Angiopoietin-2 and Podoplanin in Lymphatic Endothelial Cells on HCC Outcomes with Tyrosine Kinase Inhibitor Treatment According to Sex

**DOI:** 10.3390/biomedicines12071424

**Published:** 2024-06-26

**Authors:** Simone Lasagni, Rosina Maria Critelli, Fabiola Milosa, Dario Saltini, Filippo Schepis, Adriana Romanzi, Francesco Dituri, Grazia Serino, Lorenza Di Marco, Alessandra Pivetti, Filippo Scianò, Gianluigi Giannelli, Erica Villa

**Affiliations:** 1Gastroenterology Unit, Chimomo Department, University of Modena and Reggio Emilia, 41125 Modena, Italy; simone.lasagni@unimore.it (S.L.); rosinamaria.critelli@unimore.it (R.M.C.); fabiola.milosa@unimore.it (F.M.); dario.saltini@unimore.it (D.S.); fschepis@unimore.it (F.S.); adriana.romanzi@unimore.it (A.R.); 2Clinical and Experimental Medicine Program, Department of Biomedical, Metabolic and Neural Sciences, University of Modena and Reggio Emilia, 41125 Modena, Italy; lor.dimarco@gmail.com; 3National Institute of Gastroenterology “IRCCS Saverio de Bellis”, Research Hospital, 70013 Castellana Grotte, Italy; francesco.dituri@irccsdebellis.it (F.D.); grazia.serino@irccsdebellis.it (G.S.); gianluigi.giannelli@irccsdebellis.it (G.G.); 4Gastroenterology Unit, Azienda Ospedaliero-Universitaria di Modena, 41124 Modena, Italy; pivetti.alessandra@aou.mo.it (A.P.); sciano.filippo@aou.mo.it (F.S.)

**Keywords:** immunohistochemistry, angiogenesis, lymphangiogenesis, metastasis, sex

## Abstract

Hepatocellular carcinoma (HCC) is the second leading cause of cancer death worldwide. Curative treatments are available to a minority of patients, as HCC is often diagnosed at an advanced stage. For patients with unresectable and multifocal HCC, tyrosine kinase inhibitor drugs (TKIs) are the only potential treatment option. Despite extensive research, predictors of response to these therapies remain elusive. This study aimed to analyze the biological and histopathological characteristics of HCC patients treated with TKIs, focusing on angiogenesis and lymphangiogenesis. Immunohistochemistry quantified the expression of angiopoietin-2 (Ang2), lymphatic endothelial cells (LEC) podoplanin, and C-type Lectin Domain Family 2 (CLEC-2), key factors in neoangiogenesis and lymphangiogenesis. An increased expression of endothelial Ang2 and LEC podoplanin predicted a lower risk of metastasis. Female patients had significantly longer overall survival and survival on TKIs, associated with higher tumor expression of endothelial Ang2 and LEC podoplanin. Moreover, LEC podoplanin expression and a longer time on TKIs were independently correlated with improved survival on TKI therapy at Cox regression analysis. These findings suggest that endothelial Ang2 and LEC podoplanin could be potential biomarkers for predicting treatment outcomes and guiding therapeutic strategies in HCC patients treated with TKIs.

## 1. Introduction

Hepatocellular carcinoma (HCC) is the most common primary liver tumor and the second leading cause of cancer-related deaths worldwide. It exhibits a marked gender predisposition, ranking as the fifth most common cancer in men and the seventh in women [1]. Mortality rates associated with HCC are increasing for both genders. HCC typically develops in the context of advanced chronic liver disease, primarily due to hepatitis B virus (HBV), hepatitis C virus (HCV), and alcohol abuse. Additionally, metabolic dysfunction-associated steatotic liver disease (MASLD) has recently emerged as a significant risk factor.

In over 70% of cases, HCC is diagnosed at an advanced stage [2], leaving systemic treatment as the primary therapeutic option. Before the recent introduction of atezolizumab/bevacizumab and tremelimumab/durvalumab combinations, tyrosine kinase inhibitors (TKIs) like sorafenib and regorafenib were the only systemic therapies available [3]. TKIs, a class of small-molecule anticancer drugs, target tyrosine kinases—enzymes that play a crucial role in modulating growth factor signaling and promoting tumor growth through increased cell proliferation, angiogenesis, and metastasis and by inhibiting apoptosis [4,5]. Sorafenib was the first orally administered TKI drug followed by parent drug regorafenib. Sorafenib, the first orally administered TKI, targets multiple pathways including the RAF/MEK/ERK pathway and various VEGF receptors (VEGFR1, VEGFR2, and VEGFR3), as well as the platelet-derived growth factor receptor. Although it inhibits tumor growth and angiogenesis, leading to prolonged life, it cannot stop disease progression due to the development of resistance in tumor cells [6]. Lenvatinib, another TKI, targets VEGFA, VEGFC, fibroblast growth factor (FGF), and fibroblast growth factor receptors 1–4 (FGFR). It differs from sorafenib by having more potent antiangiogenic activity, particularly through its action on FGFR4, and its anti-lymphangiogenic activity [7].

TKIs play a crucial role in treating HCC, a cancer often marked by significant vascularization. Angiogenesis, the creation of new blood vessels, is vital for the growth and spread of solid tumors, including HCC [8]. This multi-step process begins with the dilation and increased permeability of existing blood vessels, a response to the vascular endothelial growth factor (VEGF). Tumor cells release pro-angiogenic factors that bind to specific receptors on the endothelial cells of pre-existing vessels. This binding subsequently activates the endothelial cells, initiating the angiogenic process [9,10]. VEGF acts as a highly specific mitogen that promotes the proliferation, migration, and angiogenesis of vascular endothelial cells via tyrosine kinase receptors. Additionally, it also increases vascular permeability [10]. In normal tissues, vascular endothelial growth factor (VEGF) helps maintain blood vessel density and facilitates nutrient transport. Within tumors, endogenous VEGF plays a pivotal role by triggering the release of angiopoietin-2 (Ang2), shifting its function from anti-angiogenic to pro-angiogenic. This shift significantly impacts tumor growth and development [11]. Angiopoietins are ligands of the receptor tyrosine kinase Tie-2, and their binding regulates the balance between vessel stabilization and remodeling. In the presence of VEGF, Ang2’s interaction with Tie-2 promotes endothelial cell migration and proliferation, facilitating angiogenesis. Conversely, in the absence of VEGF, Ang2 binding to Tie-2 destabilizes the cellular matrix, potentially leading to endothelial cell apoptosis and vascular regression [6,11]. Although Ang2 plays a role predominantly in vascular angiogenesis, it also appears to have a function in lymphatic vascular remodeling and maturation [12,13]. The involvement of lymphatic systems in hepatocellular carcinoma (HCC) has not been thoroughly explored, yet emerging evidence suggests their significant role in the disease’s progression and metastasis. Rapid tumor growth leads to dramatic alterations in lymphatic vessels, with lymphatic endothelial cells (LECs) experiencing sprouting and dilation. The interaction of tumor cells with LECs can allow them to enter and spread through the lymphatic vessels facilitating dissemination of tumor cells [13]. Among the markers indicating the presence of lymphatic vessels, podoplanin, expressed by LECs, stands out as particularly dependable. Under the influence of the prospero-related homeobox 1 (Prox1), immature LECs undergo maturation, expressing podoplanin. The interaction between podoplanin and the C-type lectin receptor (CLEC)-2 on platelets is believed to encourage tumor invasion and metastasis [14,15]. Interestingly, studies in models other than HCC, such as the mammary fat pad xenograft model of human breast cancer using MDA-MB-231cells, have shown that TKIs can target lymphangiogenesis [16]. This indicates a potential avenue for HCC treatment, emphasizing the need for further research in this area [17].

This study aimed to analyze the expression of Ang2, podoplanin, and CLEC-2 proteins to better understand the outcomes of patients with advanced hepatocellular carcinoma (HCC) undergoing TKI treatment. Additionally, the study investigated the potential role of sex as a modifier of treatment response.

## 2. Materials and Methods

### 2.1. Patients

We collected clinical and histopathological data from 23 patients diagnosed with HCC between 2018 and 2021, all of whom received systemic treatment with TKIs as first-line systemic treatment. The data collected included age, sex, age at diagnosis, time from diagnosis to death, time from sorafenib administration to death or switch to second line-treatment with regorafenib, time on regorafenib, overall time on TKIs, and biochemical data (creatinine, bilirubin, platelets, alpha-feto protein (AFP), albumin, and international normalized ratio (INR)). Response to TKIs was recorded as complete response (CR), stable disease (SD), or progressive disease (PD).

### 2.2. Immunohistochemistry

Using immunohistochemistry, we analyzed the expression levels of Ang2, podoplanin, and CLEC-2 from the paraffin-embedded HCC tissue and in the surrounding non-tumoral cirrhotic tissue taken by biopsy before starting TKIs.

After deparaffinization and rehydration, antigen unmasking was performed with 1 mM EDTA buffer, pH 8, at 98 °C for 15 min. The sections were then incubated in methanol 5% and H_2_O_2_ 1% for 5 min for blocking endogenous peroxidases; nonspecific sites were blocked using a blocking solution reagent with bovine serum albumin 3% for 30 min at room temperature. Sections were then incubated with goat anti-Ang2 primary antibody (AF623, R&D Systems, Inc., Minneapolis, MN, USA) at working dilution of 1:50 or mouse anti-podoplanin primary antibody (05463645001, Roche Diagnostics, Monza, MB, Italy), or rabbit anti-CLEC-2 primary antibody (LSB12627, LifeSpan BioSciences, Inc., Shirley, MA, USA) at working dilution of 1:50. Sections were then incubated with prediluted OmniMap anti-goat horseradish peroxidase conjugated secondary antibody (Ventana Medical Systems, Tucson, AZ, USA), or anti-mouse horseradish peroxidase conjugated secondary antibody (Ventana Medical Systems, Tucson, AZ, USA), or anti-rabbit horseradish peroxidase conjugated secondary antibody (Ventana Medical Systems, Tucson, AZ, USA) for 20 min in humidity chamber and then with detection kit reagents (ultra-view universal horseradish peroxidase multimer and diaminobenzidine (DAB) chromogen, Ventana Medical Systems, Tucson, AZ, USA) following the manufacturer’s instructions. The sections were then counterstained with hematoxylin, dehydrated, and permanently mounted for microscopic examination. Images of stained liver tissue were processed with ImageJ software (https://imagej.net.) to obtain the intensity value of DAB signal.

### 2.3. Statistical Analysis

Continuous variables were expressed as mean ± standard deviation (SD), and data were reported as counts and percentages. Continuous variables were compared with Student’s *t* test and categorical variables with Pearson’s chi-square test [18]. Bivariate regression correlation tests were used to compare all variables in both groups and to determine whether they correlated significantly with each other in determining clinical outcome of treatment or in characterizing disease aggressiveness. The nonparametric Mann–Whitney U test, however, was used to obtain additional specificity of the result when the parametric test was significant [19]. The Kruskal–Wallis independent samples test was used to observe the distribution of variables in the patients’ age quartile [20]. Univariate and multivariate Cox regression analyses were used to identify variables associated with HCC mortality. The following variables, all obtained at the time of HCC diagnosis, were evaluated in the univariate analysis: age at diagnosis, sex, etiology, varices, PVT, HCC grade, extra-hepatic localization, time on TKIs, tumor Ang2, endothelial Ang2, endothelial podoplanin, tumoral CLEC-2, bilirubin, INR, albumin, creatinine, platelet, and AFP. Variables with a *p*-value < 0.10 at univariate analysis were included in the multivariate models. To prevent a high level of interaction between the different variables, these were tested for collinearity, and we excluded the collinear variables from the multivariate model, predicting the clinical course of TKI-treated HCC.

To visualize the capacity of the histochemical features to discriminate between HCCs developing extra-hepatic localization, we summarized the data in a receiver operating characteristic curve [21]. The Kaplan–Meier method was used to estimate the cumulative probability of overall survival. Patients were censored at the time of LT, death, or last visit. Differences in observed probability were assessed using the log-rank test.

The study protocol was approved by the Ethics Committee of the Emilia Vasta Nord Area (AVEN), (239/12_CE_UniRer_MO).

PASW Statistics (ver. 28; IBM Corporation, Armonk, NY, USA) was used for statistical analysis.

## 3. Results

### 3.1. Clinical Results

The demographic, clinical, and biochemical characteristics of the study cohort, along with a breakdown by sex, are detailed in Table 1. At the time of HCC diagnosis and the initiation of TKI therapy, women were slightly but significantly older than men.

Patients received sorafenib as their first-line systemic therapy and regorafenib as their second-line therapy. Two patients (females) showed CR, 9 patients (7 males and 2 females, 24.1% and 33.3%, respectively) showed stable disease (SD), and 24 patients (22 males and 2 females, 75.9% and 33.3%, respectively) had PD (*p* = 0.004, chi-square test). During the follow-up period, 31 patients (88.6%) died, with a slight trend toward lower mortality among women (Table 1). Notably, the time from diagnosis of HCC to death and the duration from the initiation of sorafenib treatment to death were both significantly longer in women (Table 1). This was confirmed as a Kaplan–Meier analysis that showed a survival advantage for women both for survival time from diagnosis (Figure 1A) and for survival on TKIs (Figure 1B). 

### 3.2. Immunohistochemical Results

Ang2 expression was observed in the parenchymal and endothelial cells of both male and female patients. Specifically, cytoplasmic expression was noted in the tumor parenchyma, while the nuclei were devoid of expression. Additionally, endothelial expression of Ang2 was detected in hepatic sinusoids and the endothelia of connective tissue surrounding the tumor (Figure 2 and Table 2). This revealed variations in staining intensity, with female patients showing higher expression levels. In contrast, podoplanin expression was confined exclusively to the lymphatic endothelia, with no expression detected in the cytoplasm or nuclei of hepatocytes. Interestingly, within this context, females also demonstrated higher expression levels of podoplanin compared to males (Figure 2 and Table 2).

Conversely, CLEC-2 staining was primarily observed in the cytoplasm of hepatocytes, with minimal nuclear presence, and was absent in endothelial cells.

Levels of Ang2 expression in endothelial cells were notably higher in patients older than 65 years (65 years older vs. younger than 65 years: 0.576 ± 0.084 vs. 0.487 ± 0.061, OD (*p* = 0.012, Mann–Whitney U test)). The distribution of endothelial Ang2 was significantly different within the age quartiles (*p* = 0.014; Kruskal–Wallis test). Interestingly, all women were clustered in the third age quartile. A similar relationship with age was observed in the podoplanin expression in LEC (patients younger than 65 vs. those older than 65: 0.645 ± 0.109 vs. 0.760 ± 0.090 (*p* = 0.012, Mann–Whitney U test)).

A receiver operating characteristic (ROC) curve analysis to evaluate the diagnostic power for predicting extra-hepatic spread of the two significantly more expressed proteins showed that the AUC values were 0.839 (95% CI 0.679 to 0.999) and 0.821 (95% CI 0.652 to 0.989) (*p* < 0.0001) for LEC podoplanin and endothelial Ang2, respectively (Figure 3). For both proteins, lower levels were predictive of a higher risk of extra-hepatic dissemination.

#### Cox Regression Analysis

Univariate Cox analysis for survival identified endothelial podoplanin expression, time on TKIs, and presence of extra-hepatic tumor localizations as significantly related with survival (Table 3). Endothelial podoplanin expression and time on TKIs were independently related with survival. Extra-hepatic localization was not included in the model as collinear with the other two factors.

Univariate Cox analysis for prediction of extra-hepatic localization showed that younger age at diagnosis, shorter time on TKIs, lower endothelial Ang2 expression, and lower LEC podoplanin expression were significantly related with a higher risk of extra-hepatic spread. However, as LEC podoplanin expression was collinear with all three other factors, the multivariable model was not built (Appendix A).

## 4. Discussion

In this study, we evaluated patients with HCC undergoing treatment with TKIs to assess the relationship between the expression of Ang2, podoplanin, and CLEC-2 and outcomes of TKI therapy. This investigation explores the interplay among angiogenesis, lymphangiogenesis, and HCC progression in response to TKI treatment. TKIs are broad-spectrum inhibitors that target multiple kinases. Specifically, sorafenib blocks the RAF/MEK/ERK pathway and several receptors, including VEGFR1, VEGFR2, and VEGFR3, as well as the platelet-derived growth factor receptor [4,5,6,7]. VEGFR3, a receptor for VEGF-C, plays a critical role in lymphangiogenesis and is involved in both normal and pathological lymphangiogenesis across various cancer types [22].

VEGF-D binds to both VEGFR2 and VEGFR3, promoting both angiogenesis and lymphangiogenesis [22]. Although less studied than angiogenesis, lymphangiogenesis significantly influences hepatocellular carcinoma (HCC) development and progression. These pathways are distinct yet interconnected, with the lymphatic system playing a crucial role in tumor spread [23]. Data from various solid cancers consistently show a higher risk of metastasis in patients with increased lymphangiogenesis [24], which appears to contradict our findings, which show a better outcome on TKI treatment for patients with higher levels of podoplanin. However, our focus is on a very specific aspect, namely the response to a drug that targets these mechanisms. TKIs, and specifically sorafenib, inhibit all three vascular endothelial growth factors, including VEGF-C, providing crucial insight into our results. It is clear that these patients, who were only eligible for systemic therapy after failing more curative treatments, were at an advanced stage with expected extra-hepatic spread, which aligns with high podoplanin levels. From this perspective, the elevated endothelial Ang2 levels also indicate aggressive disease [25,26]. Notably, females, showing the highest podoplanin and Ang2 levels, responded better to TKI therapy, had longer overall survival, and remained on treatment longer compared to males, who exhibited lower levels of these markers.

Overall, female patients achieved more favorable outcomes, achieving stable or complete responses to TKI treatment more often than their male counterparts, despite being older and having more advanced disease at the beginning of the study. This is particularly noteworthy because all the women in the study were post-menopausal. Typically, females develop hepatocellular carcinoma (HCC) at a later age than males. This phenomenon is associated with significant systemic and hepatic changes toward a pro-inflammatory state, characterized by elevated levels of IL-6 and TNF-α in both the circulation and the liver [27,28]. These changes result in a distinct shift in the hepatic microenvironment that ultimately promotes carcinogenesis [13,14,29]. The increase in these inflammatory factors, driven by menopause, induces a systemic and hepatic pro-inflammatory state, creating a conducive environment for tumor development. Moreover, as women enter menopause, they rapidly transition from an estrogen-protected environment to one lacking this hormonal protection, where inflammation is less controlled, thereby increasing the risk of initiating processes leading to HCC development. It is important to note that chronic inflammation is among the stimuli that can activate lymphangiogenesis [14]. The activation of VEGFR-3 by its ligand, VEGF-C, is the principal signaling pathway activated during chronic inflammation [30,31]. The altered environment created by menopause in women could provide an ideal setting for triggering lymphangiogenesis, further explaining the observed therapeutic responses. Further evidence supporting this view comes from the role played by estrogens on lymphangiogenesis. Several studies [32,33] indicate that estrogens inhibit lymphatic growth while simultaneously stimulating vascular endothelial cells to proliferate and form capillaries [32]. The decline in estradiol levels at menopause, or the blocking of estradiol effects by tamoxifen could pave the way for uncontrolled growth of lymphatics [33]. Intriguingly, in the context of clear cell renal cell carcinoma (a sex-biased tumor) [34], a higher androgen receptor expression was associated with decreased lymphatic metastases. This suggests an influence on lymphangiogenesis that is opposite to that observed in females [35].

The relationship between podoplanin expression and survival in HCC patients treated with TKIs, as evidenced by multivariate Cox regression analysis, was notably strong. LEC podoplanin and duration of TKI treatment emerged as the only independent prognostic factors. Additional factors identified by Cox regression analysis that predicted extra-hepatic spread included age at diagnosis and endothelial Ang2. However, due to collinearity among these factors, a multivariate model was not feasible. Nevertheless, the significance of these factors in predicting outcomes that heavily influence prognosis cannot be overlooked.

In conclusion, our study demonstrates that the endothelial expression of Ang2 and podoplanin can differentiate patients likely to exhibit more aggressive and advanced HCC, particularly in females where normal hormonal protection diminishes with age. Podoplanin, in particular, has emerged as a predictor of the clinical course of HCC. Female patients with higher levels of this protein in the lymphatic endothelium were more likely to respond to therapy and thus had longer survival times. These proteins, significant markers of tumor biology, are crucial in selecting the most appropriate treatment to improve prognosis, which would otherwise be extremely poor. Our findings support previous studies that have shown better outcomes for women on TKI therapy [36,37] and suggest a potential mechanism for this more favorable outcome.

## Figures and Tables

**Figure 1 biomedicines-12-01424-f001:**
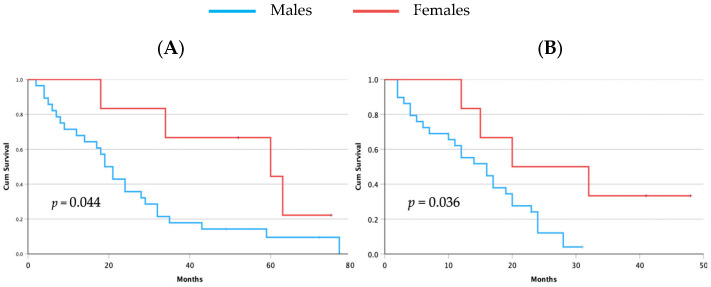
Kaplan–Meier analysis of overall survival from time of HCC diagnosis to death (**A**) and survival time from starting TKIs to death (**B**). For both survival analyses, females showed a significantly longer survival. Most importantly, females showed a significantly higher survival on TKIs.

**Figure 2 biomedicines-12-01424-f002:**
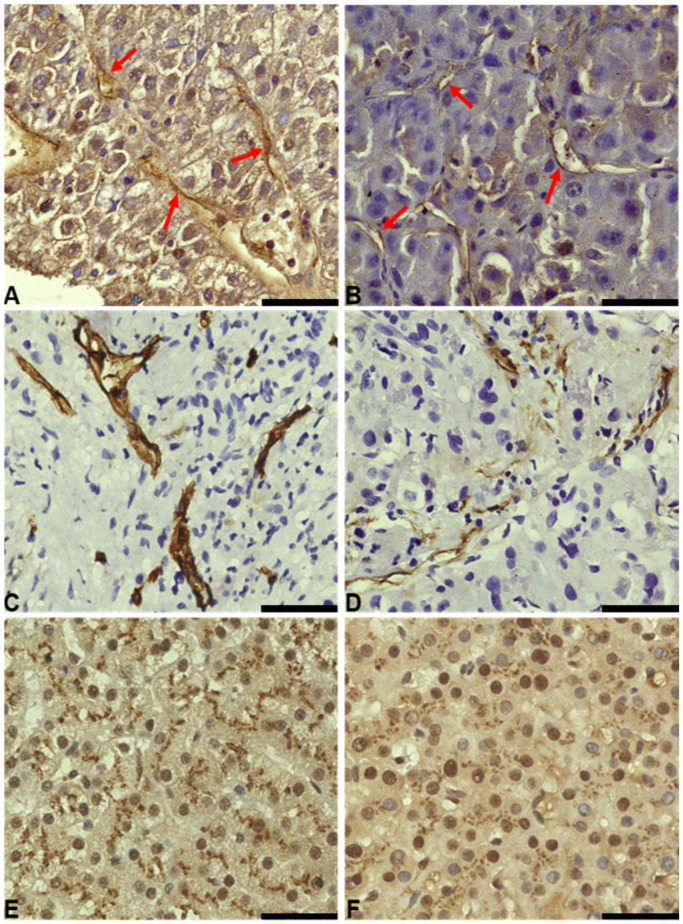
Comparison of Ang2, podoplanin, and CLEC-2 staining in female and male patients treated with TKIs (50 µm scale bar). (**A**) 63× image of Ang2 staining in tumor and endothelial tissue (red arrows for endothelia) of a female patient. (**B**) 63× image of Ang2 staining in tumor and endothelial tissue (red arrows for endothelia) of a male patient. (**C**) 63× image of podoplanin staining at the level of the lymphatic endothelium in a female patient. (**D**) 63× image of podoplanin staining at the level of the lymphatic endothelium in a male patient. (**E**) 63× image of CLEC-2 staining at the level of tumor tissue in a female patient. (**F**) 63× image of CLEC-2 staining at the level of tumor tissue in a male patient.

**Figure 3 biomedicines-12-01424-f003:**
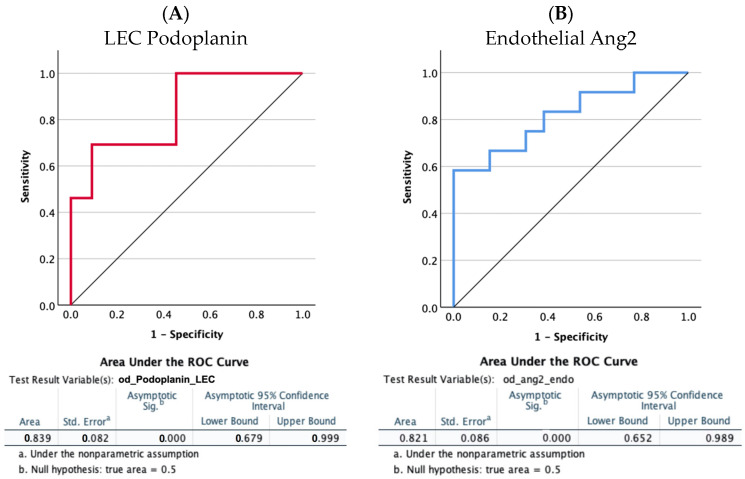
Receiver operating characteristic (ROC) curve analysis to evaluate the diagnostic power of the LEC podoplanin (**A**) and of endothelial Ang2 (**B**) to predict extra-hepatic spread. The area under the ROC curves (AUCs) was analyzed using the Hanley and McNeil method [21]. The AUC values were 0.839 (95% CI 0.679 to 0.999; *p* < 0.0001) and 0.821 (95% CI 0.652 to 0.989; *p* < 0.0001).

**Table 1 biomedicines-12-01424-t001:** Baseline demographic and biochemical parameters in patients treated with TKIs, analyzing differences in relation to sex using the chi-square test or the Mann–Whitney test as appropriate.

	Whole Cohort (*n* = 35) (%)	M (*n* = 29) (%)	F (*n* = 6) (%)	*p*
Age at diagnosis (M ± SD)	65.0 ± 9.6	63.5 ± 9.9	72.3 ± 3.4	0.04
Age at TKI start (M ± SD)	66.4 ± 9.9	64.8 ± 10.1	73.8 ± 4.3	0.04
Deaths	31 (88.6)	27 (93.1)	4 (66.7)	0.06
Etiology				0.46
HCV	17 (48.6)	13 (44.8)	4 (66.7)	
HBV	5 (14.3)	5 (17.2)	0	
Dysmetabolic	13 (37.1)	11 (37.9)	2 (33.3)	
Child–Pugh class (A or B)	30, 5	27, 2	3, 3	0.06
HCC grade (1, 2, or 3)	10, 16, 9	8, 13, 8	2, 3, 1	0.98
Extra-hepatic localization	16 (55.2)	16 (55.2)	0	0.01
Site of extra-hepatic localization				
Lung	8	8	0	
Bone	4	4	0	
Lymph nodes	4	4	0	
Peritoneum	4	4	0	
Time HCC diagnosis–death (months) (M ± SD)	37.7 ± 28.6, 32	27.5 ± 24.0	47.5 ± 19.9	0.44
Time TKIs–death (months) (M ± SD)	16.8 ± 11.0, 16	14.6 ± 8.9	27.8 ± 14.3	0.44
TKI response (CR, SD, or PD)	2, 9, 24	0, 7, 22	2, 2, 2	0.04
Overall time on TKIs	11.1 ± 10.9	9.1 ± 8.4	21.1 ± 16.5	0.07
Bilirubin (mg/dL) (M ± SD)	1.4 ± 1.6, 0.93	1.9 ± 2.5	1.2 ± 0.8	0.96
Albumin (g/dL) (M ± SD)	3.9 ± 0.46, 4.00	3.9 ± 0.5	4.1 ± 0.07	0.40
INR (M ± SD, median)	1.13 ± 0.18, 1.09	1.1 ± 0.1	1.1 ± 0.1	0.87
Creatinine (mg/dL) (M ± SD)	1.02 ± 0.33, 0.91	1.0 ± 0.3	1.0 ± 0.2	0.68
Platelets (109/L) (M ± SD)	148 ± 101	148 ± 124	118 ± 18	0.85
AFP (ng/mL) (M ± SD)	6634 ± 17,984	4148 ± 13,755	30,252 ± 42,777	0.77

**Table 2 biomedicines-12-01424-t002:** Ang2, podoplanin, and CLEC-2 staining intensity (expressed as optical density (OD)) in HCC patients treated with TKIs stratified according to sex.

Protein	Females (*n* = 5)	Males (*n* = 18)	*p*
Ang2			
Endothelial	0.623 ± 0.09	0.514 ± 0.07	0.008
Hepatocyte (T)	0.527 ± 0.15	0.468 ± 0.10	0.316
Podoplanin			
Endothelial	0.800 ± 0.07	0.680 ± 0.11	0.036
CLEC-2			
Hepatocyte (T)	0.508 ± 0.06	0.597 ± 0.14	0.204

(Ang2, angiopoietin-2; T, tumor; CLEC-2, C-type lectin receptor).

**Table 3 biomedicines-12-01424-t003:** Univariate and multivariate Cox regression analysis for survival.

	Univariate Analysis		Multivariate Analysis	
Variables	HR (95% CI)	*p*	HR (95% CI)	*p*
Age at diagnosis	0.978 (0.941–1.016)	0.260		
Sex	0.425 (0.146–1.233)	0.115		
Etiology	0.869 (0.586–1.287)	0.483		
Varices	1.142 (0.455–2.868)	0.777		
PVT	1.346 (0.568–3.186)	0.500		
HCC grade	1.141 (0.736–1.769)	0.556		
Extra-hepatic localization *	1.982 (0.928–4.237)	0.077		
Time on TKIs	0.950 (0.911–0.990)	0.015	0.944 (0.893–0.998)	0.043
Tumoral Ang2	1.334 (0.025–71.640)	0.887		
Endothelial Ang2	0.253 (0.002–35.210)	0.586		
LEC podoplanin *	0.027 (0.001–0.876)	0.042	0.009 (0.000–0.842)	0.042
Tumoral CLEC-2	0.641 (0.009–47.284)	0.840		
Bilirubin	0.835 (0.585–1.191)	0.320		
INR	0.637 (0.041–9.826)	0.747		
Albumin	0.762 (0.290–2.007)	0.583		
Creatinine	0.970 (0.257–3.657)	0.965		
Platelets	1.000 (1.000–1.000)	0.189		
AFP	1.000 (1.000–1.000)	0.408		

* Collinear with LEC podoplanin.

## Data Availability

The data presented in this study are available on request from the corresponding author. The data are not publicly available due to privacy reasons.

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
