# Peer review of "Differential Impact of Tumor Endothelial Angiopoietin-2 and Podoplanin in Lymphatic Endothelial Cells on HCC Outcomes with Tyrosine Kinase Inhibitor Treatment According to Sex"

_biomedicines, 2024, doi:10.3390/biomedicines12071424_

Round 1
Reviewer 1 Report
Comments and Suggestions for Authors
Summary
The article submitted by the authors is entitled, “Impact of Tumor Endothelial Angiopoietin-2 and Lymphatic Endothelial Cells Podoplanin on HCC Outcomes with Tyrosine Kinase Inhibitor Treatment” The work is clear and concise, effectively supported by the original research paper, reviews, and additional literature. In this manuscript, the authors have done immunohistochemistry of the patient and overall survival by giving the TKI (Tyrosine Kinase Inhibitor). This study is supported by various statistical tests such as the Mann-Whitney U test, the Kruskal-Wallis independent-samples test for observing the distribution of the variable in the patients' age quartiles, and Cox regression analysis to identify the variables associated with HCC. However, this manuscript needs minor corrections before publication.
Comment:
1. The title of the paper does not convey the author's research objectives. However, the conclusion of the paper indicates that the study focuses on a comparative analysis of the effects of TKIs on male and female HCC patients.
2. Referencing should be done in the proper format, doi is not available for reference no 3,5,6,7,8,9,11,14,15, 27, etc.
3. Grammatical errors should be removed thoroughly.
4. The author should add more references related to statistical analysis for a better understanding of the statistical tests such as the Mann-Whitney test, chi-square test, and Kruskal-Wallis independent test.
Comments on the Quality of English LanguageMinor editing of English language is required.
Author Response
Comment:
- The title of the paper does not convey the author's research objectives. However, the conclusion of the paper indicates that the study focuses on a comparative analysis of the effects of TKIs on male and female HCC patients.
We acknowledge Reviewer #1 comment. We have modified the title accordingly.
- Referencing should be done in the proper format, doi is not available for reference no 3,5,6,7,8,9,11,14,15, 27, etc.
We have added and highlighted in red the missing doi. Reference 14 and 15 do not have a doi, as Anticancer Research started late to have doi. Please take into account that the numbers of the above cited from ref 17 on have changed.
- Grammatical errors should be removed thoroughly.
We have thoroughly revised style and grammar.
- The author should add more references related to statistical analysis for a better understanding of the statistical tests such as the Mann-Whitney test, chi-square test, and Kruskal-Wallis independent test.
As required we have added the references for the above indicated statistical tests.
Comments on the Quality of English Language
Minor editing of English language is required.
We have thoroughly revised the whole manuscript. Changes are highlighted in red.
Reviewer 2 Report
Comments and Suggestions for Authors
In this manuscript, the author analysis the expression of Ang2, podoplanin and CLEC-2 proteins to understand the outcomes of patients with HCC undergoing TKI treatment. This study demonstrates that endothelial expression of Ang2 and podoplanin can discriminate patients likely to present with more aggressive and advanced HCC, particularly in females where normal hormonal protection diminishes with age. Specifically, podoplanin has been identified as a predictor of the clinical course of HCC. In my opinion, the manuscript could be published on Biomedicines after revision:
1. The ethics approval information of this study should be stated in the paper and uploaded as an attachment;
2. The format of the table needs to be carefully adjusted;
3. Much more discussion of these result, especially for the difference between male and female, should be added in this manuscript;
Author Response
- The ethics approval information of this study should be stated in the paper and uploaded as an attachment;
The ethics approval had been already indicated on page 10, lines 344-345. We have uploaded the the
pdf of the approval as attachment, as indicated.
- The format of the table needs to be carefully adjusted
We have carefully reviewed the tables.
3. Much more discussion of these result, especially for the difference between male and female, should be added in this manuscript;
We have added a paragraph on the specific effects on lymphangiogenesis of female and male hormones (Lines 316-324).